# Targeting Heat-Shock Protein 90 in Cancer: An Update on Combination Therapy

**DOI:** 10.3390/cells11162556

**Published:** 2022-08-17

**Authors:** Xiude Ren, Tao Li, Wei Zhang, Xuejun Yang

**Affiliations:** 1Department of Neurosurgery, Tianjin Medical University General Hospital, Tianjin 300052, China; 2Laboratory of Neuro-Oncology, Tianjin Neurological Institute, Tianjin 300052, China; 3Departments of Cancer Biology, Wake Forest School of Medicine, Winston-Salem, NC 27157, USA; 4Wake Forest Baptist Comprehensive Cancer Center, Winston-Salem, NC 27157, USA; 5Department of Neurosurgery, Tsinghua University Beijing Tsinghua Changgung Hospital, Beijing 102218, China

**Keywords:** heat-shock protein 90 inhibitors, molecular chaperones, HSP90, cancer therapy, combinational therapy

## Abstract

Heat-shock protein 90 (HSP90) is an important molecule chaperone associated with tumorigenesis and malignancy. HSP90 is involved in the folding and maturation of a wide range of oncogenic clients, including diverse kinases, transcription factors and oncogenic fusion proteins. Therefore, it could be argued that HSP90 facilitates the malignant behaviors of cancer cells, such as uncontrolled proliferation, chemo/radiotherapy resistance and immune evasion. The extensive associations between HSP90 and tumorigenesis indicate substantial therapeutic potential, and many HSP90 inhibitors have been developed. However, due to HSP90 inhibitor toxicity and limited efficiency, none have been approved for clinical use as single agents. Recent results suggest that combining HSP90 inhibitors with other anticancer therapies might be a more advisable strategy. This review illustrates the role of HSP90 in cancer biology and discusses the therapeutic value of Hsp90 inhibitors as complements to current anticancer therapies.

## 1. Introduction

The HSP family is a group of molecular chaperones facilitating the folding and maturation of a group of proteins termed “client” proteins. When exposed to environmental stressors (heat, heavy metals, hypoxia, and acidosis), cells increase the expression of HSPs as an adaptive response to maintain cell homeostasis [1]. Based on molecular weights, the HSP family can be classified into HSP27, HSP40, HSP60, HSP70, HSP90, and large HSPs. Members of this family are closely linked to each other through a signal network [2]. This review will mainly focus on HSP90.

HSP90 participates in client protein cycles differently from other chaperones. In contrast to Hsp70 and Hsp60, HSP90 does not facilitate the folding of newly synthesized proteins but mainly plays an important role in the late-stage maturation, activation, and stability of client proteins [3]. Furthermore, HSP90 interacts with a more specific client group that is highly dependent on HSP90 for activation. The HSP90 clients mostly comprises unstable signaling molecules, particularly kinases and transcription factors regulating signal transduction, protein trafficking, chromatin remodeling, autophagy, cell proliferation, and survival [1,4]. Many of these clients are oncogenic proteins considered key drivers of the occurrence and development of cancers.

Many malignant behaviors, including tumor growth, adhesion, invasion, metastasis, and angiogenesis, are inseparable from HSP90 regulation and protection [2]. HSP90’s expression level is extremely high in various malignancies, indicating that cancer cells are under increased pressure to maintain protein integrity and hence have a higher demand for HSP90 [5]. In addition, overexpression of HSP90 in tumor tissues is often associated with poor prognosis [6,7]. Hsp90 inhibition can lead to the degradation of oncogenic proteins and disruption of oncogenic pathways [8]. Overall, HSP90 is an ideal therapeutic target in cancer.

The HSP90 molecule is a homodimer composed of two monomers. The HSP90 monomer contains four domains, which are all important for HSP90’s functions. In addition, HSP90’s normal functions are driven by its ATPase activity, binding and hydrolyzing ATP. Many HSP90 inhibitors have been developed based on the structure and function of HSP90; these inhibitors have shown anticancer activity in many preclinical trials. However, the outcomes of HSP90 inhibitor monotherapy were not as good as expected. Toxicity is the main factor that limits the efficacy of HSP90 inhibitors as monotherapies [9], along with tumor resistance to the inhibitors.

Combining HSP90 inhibitors with other anticancer therapies is a promising strategy to address this dilemma. Many preclinical and clinical studies have shown HSP90 inhibitors’ synergistic effects with other anticancer therapies, such as chemotherapy and radiotherapy. Here, we review the biological function of HSP90 in cancer and the therapeutic value of Hsp90 inhibitors in combination therapy.

## 2. Structure and Function of HSP90

HSP90 exists in cells in the form of a homodimer. Each monomer comprises four domains: the N-terminal domain (NTD), the middle domain (MD), the C-terminal domain (CTD), and a charged linker region of negative charges connecting the N-terminal and middle domains (Figure 1). The sequence of the charged linker can influence HSP90’s functions [10].

The NTD is responsible for the binding of ATP with an adenine nucleotide-binding pocket at the N-terminus. X-ray crystallography studies revealed that this binding is closely related to the Bergerat fold in the ATP binding pocket [11]. With its ATPase activity, the NTD functions as a driver of certain alterations such as the binding of client proteins. The MD is also reported to modulate HSP90 ATPase activity and binding to substrates [12]. The CTD is important for HSP90 dimerization, which is the structural basis for the normal functioning of HSP90. The CTD adenine nucleotide-binding pocket activates only when the N-terminal site is occupied [13]. The CTD is also the binding site for small molecules such as nucleotides, novobiocin, and cisplatin [1].

There are four isoforms of HSP90: HSP90α (inducible form), HSP90β (constitutive form), GRP94 (glucose-regulated protein 94), and TRAP1 (tumor necrosis factor receptor-associated protein 1). HSP90α and HSP90β share a similar structure, function, and intracellular localization in the cytoplasm. GRP94 is a glycoprotein that can protect cells from endoplasmic reticulum stress. GRP94 has a special client group, including secreted and membrane proteins including the immunoglobulin (Ig) family, Toll-like receptor, and integrins [14]. Like the cytosolic isoforms, GRP94 also contains NTD, MD, CTD and the charged linker TRAP1 is a mitochondrial isoform of HSP90 mostly located in the mitochondrial matrix. Unlike other isoforms, TRAP1 does not have the charged linker domain. TRAP1 is essential for mitochondrial homeostasis in some pathological status [15]. HSP90’s normal function requires assistance from other components, and cochaperones are the most important. Cochaperones are proteins that aid HSPs in conformational cycling and the binding of substrates. Some cochaperones containing tetratricopeptide repeat (TPR) domains interact with HSP90 by binding to its C-terminal MEEVD motif. Among the most specific TPR-containing cochaperones is HSC70/HSP90-organizing protein (HOP), which can bind to both HSP90 and HSP70. HOP functions as a stabilizer of the HSP90 open conformation [4]. Other TPR-containing cochaperones include Tah1, protein phosphatase 5 (PP5), CHIP, cyclophilin 40 (CYP40), and tetratricopeptide repeat domain 4 (TTC4) [16]. In addition to that, there are also non-TPR-containing cochaperones. A well-studied non-TPR-containing cochaperone, cell division cycle 37 (Cdc37), is reported to be specific for the maturation of kinases and leads to partial inhibition of HSP90 ATPase activity [17]. Cdc37 is associated with tumor formation and progression. Other non-TPR-containing cochaperones include the Activator of HSP90 ATPase homolog 1 (activating ATPase activity of HSP90) and p23 (inhibiting HSP90 ATPase activity) [18].

## 3. HSP90 Clients

The range of HSP90 clients is more limited than that of other molecular chaperones, such as HSP70 or HSP60, which interact with all unfolded proteins. HSP90s are not considered a facilitator of the folding of newly synthesized proteins but a player mediating late steps in protein maturation and maintaining metastable proteins in activatable states [18]. Therefore, HSP90 clients are proteins that primarily require HSP90’s involvement for maturation and maintaining structural stability.

In general, the known HSP90 client proteins can be classified into three categories: protein kinases such as c-SRC/v-SRC, ERBB2, EGFR, c-RAF-1/v-RAF-1, BRAF, AKT (PKB), MET, and MOK; transcription factors such as HIF-1 and p53; and structurally unrelated clients such as hTERT [19]. Detailed information on HSP90’s clients can be found at (http://www.picard.ch/downloads/Hsp90facts.pdf) (Accessed on 16 August 2022). These clients play key regulatory roles in many physiological and biochemical processes in the cell. Some clients are also key oncogenic proteins for the growth and/or survival of cancer cells. We list a fraction of them in Table 1.

It is worth mentioning that some client proteins are still under HSP90’s modulation even in mutant forms, such as V600E B-Raf [26], BCR-FGFR1 [27] and FGFR3-TACC3 [28]. These mutant gene proteins are particularly dependent on the chaperone function of HSP90. HSP90 can stabilize their structure, promote their activation, and assist them in evading proteasomal degradation [27]. HSP90 plays an important role in the development of cancer driven by mutated fusion genes.

## 4. Roles of HSP90 in Cancer

Tumor cells are subject to multiple stressors, including altered signal pathways, mutated client proteins, hypoxia, low pH, and extra demands for nutrition. Tumor cells are highly dependent on oncoproteins involved in maintaining internal homeostasis when exposed to these hostile stressors. Reduced expression of oncoproteins or increased depletion is more lethal in tumor tissues than in normal cells. Many oncoproteins are expressed in mutated forms; therefore, their metastability and activation are more dependent on HSP90 chaperones [29]. Research on *D. melanogaster* and *Arabidopsis thaliana* model systems has demonstrated that HSP90 can function as a biochemical buffer conferring cancer cells an ability to tolerate mutated proteins and altered signal pathways [1]. Therefore, higher levels of HSP90 permit oncogenic proteins to fold properly in a hostile environment. This overwhelming dependence of tumor cells on HSP90 is among the reasons why HSP90 is considered an ideal anticancer target [19].

HSP90 is important for multiple steps in malignant transformation and progression, including tumor proliferation, migration, invasion, antiapoptosis, immortalization, angiogenesis, and therapeutic resistance (Figure 2). For example, abnormal telomerase activity is observed in most human cancers but lacking in normal cells. In immortalized tumor cells, HSP90 interacts with the hTERT (human telomerase reverse transcriptase) promoter, and inhibition of HSP90 can result in decreased hTERT expression [30]. The serine/threonine kinase AKT (also known as PKB) controls key cellular processes such as proliferation and antiapoptotic effects. Inhibition of HSP90 downregulates the expression of Akt kinase and, as a result, sensitizes tumor cells to proapoptotic factors [31,32]. Focal adhesion kinase (FAK) and integrin-linked kinase (ILK) are critical promoters to cell adhesion. Inhibition of HSP90 can induce the depletion of FAK and ILK in cancer cells [33,34]. HIF-1α overexpression is reportedly linked to angiogenesis and antiapoptosis effects in cancer cells. HSP90 inhibitors can reverse the overexpression of HIF-1α (hypoxia inducible factors-1α) and simultaneously downregulate survival signaling pathways [35]. Human vascular endothelial cell growth factor (VEGF) is known to be a key player in angiogenesis via the HIF-1α/VEGF/VEGFR-2 signaling pathway. Blocking the VEGF-related pathway with an HSP90 inhibitor can suppress angiogenesis in breast cancer [36].

HSP90 is also closely related to tumor treatment resistance. For instance, therapeutic resistance to some DNA-targeted approaches is a consistent obstacle to cancer treatment. When tumor cells are exposed to treatments such as ionizing radiation or alkylation agents, the HSP family is the first line of defense to maintain DNA integrity and cell integrity [37].

In the past few years, the role of HSP90 in the DNA repair pathway has been gradually revealed with the in-depth study of the HSP90 client group. DNA damage is mediated by members of the phosphoinositide 3-kinase (PI3K)-related kinase (PIKK) family, comprising ATR (Rad3-related), ataxia telangiectasia mutated (ATM), and DNA-PKcs (DNA-dependent protein kinase catalytic subunit). This trinity of key kinases constitutes the core of the DNA damage response [38]. When double-strand breaks (DSBs) occur, MRE11-RAD50-NBS1 (MRN), a complex that can sense DSBs, recruits HSP90 to the repair foci and activates ATM with HSP90’s assistance [37]. In addition, Quanz et al. found that HSP90 is rapidly phosphorylated when DNA damage occurs, and this phosphorylation is mainly dependent on DNA-PK and ATM. Subsequently, phosphorylated HSP90 accumulates at the site of damage as a key promoter of repair [39]. Moreover, DNA-PK and ATR are identified as client proteins [40,41]. Therefore, pharmacological inhibitors of HSP90 can interfere with the DNA damage response by mediating the degradation of related proteins. Numerous in vitro and in vivo experiments have shown that HSP90 inhibitors can be used as tumor radiation sensitizers to enhance the killing effect on tumor tissues [35,42,43]. The complete mechanism by which HSP90 participates in the DNA damage repair pathway has not been entirely revealed, but the key roles of HSP90 in DNA damage repair provide perspectives for addressing the treatment resistance of some anticancer therapies.

Intriguingly, HSP90 plays a supportive role in the immune response against cancer. First, both intracellular and extracellular Hsp90 are involved in the process of antigen presentation. On the one hand, intracellular HSP90 is responsible for escorting antigenic peptides to TAP1/2 (transporters associated with antigen processing 1/2) and then into the ER (endoplasmic reticulum). In the ER, antigenic peptides are loaded onto newly synthesized MHC-I molecules. Eventually, MHC-I molecules carrying antigenic peptides are transported to the surface of tumor cells and are recognized by CD8+ T cells. On the other hand, extracellular Hsp90 secreted in the form of exosomes can bind to peptide antigens in the extracellular matrix. Then, HSP90 plus peptide antigens are recognized by HSP receptors on antigen-presenting cells (APCs) and internalized to be degraded by the proteasome. These processed peptide antigens are loaded onto MHC-II molecules in the ER and transported to the surface of APCs. Eventually, this leads to the activation of CD4+ T cells [33]. Notably, the role of HSP90 in antigen presentation has led to the research and development of HSP90-based cancer vaccines, which are not covered in detail here due to a lack of space [44].

In addition, extracellular Hsp90 is thought to be a signal for danger/damage-associated molecular patterns (DAMPs). Extracellular Hsp90 can promote the secretion of activating cytokines (IL-12) and the expression of costimulatory molecules, which potently stimulate T cells [34]. Extracellular Hsp90 also assists in the folding of receptors on immune cells such as natural killer cells and T lymphocytes [33].

## 5. HSP90 Inhibitors

As stated above, the HSP90 monomer is composed of four domains: the N-terminal domain (NTD), the middle domain (MD), the C-terminal domain (CTD) and a charged linker region. Based on the molecular structure of HSP90, HSP90 inhibitors can be divided into two categories: N-terminal inhibitors and C-terminal inhibitors.

### 5.1. The N-Terminal Inhibitors

Inhibiting HSP90 pharmacologically began with the discovery of the natural products geldanamycin (GA) (Figure 3) and radicicol (RD) (Figure 4). GA and RD were successively found to prevent ATP binding and hydrolysis of HSP90 by binding to the N-terminal ATP binding site. As a result, HSP90s with impaired conformational alterations cannot bind to the client proteins, eventually leading to the ubiquitin-mediated proteasomal degradation of clients [45]. Although GA displayed potent anticancer activity both in vitro and in vivo, its poor solubility and hepatotoxicity limit its use in clinical treatment. RD also has strong in vitro cytotoxic effects but is ineffective in vivo because of its structural instability. Although these two compounds cannot be incorporated into clinical treatment, they provide the basis for the future introduction of HSP90 inhibitors into clinical practice and serve as chemical probes to explore the functions of HSP90 [8].

Overcoming poor solubility and toxic side effects at therapeutic doses is an important concern in HSP90 inhibitor development. Various synthetic derivatives have since been developed. The modified geldanamycin derivatives 17-AAG (Figure 3) and 17-DMAG (Figure 3) are two major first-generation HSP90 inhibitors. The poor solubility of 17-AAG limits its application. Although 17-DMAG demonstrated significant anticancer activity with better solubility, toxic side effects remained [46]. IPI-504 (retaspimycin hydrochloride) (Figure 3), another derivative of GA with better water solubility, was effective and well-tolerated in advanced gastrointestinal stromal tumors (GISTs) or other soft-tissue sarcomas (STSs) [47]. WK88-1 is also a GA derivative with significant anticancer activity and much lower toxicity [48].

With the progress of academic research and the improvement of industry, pharmacological companies have developed small-molecule synthetic HSP90 inhibitors referred to as second-generation HSP90 inhibitors. Compared to the first-generation HSP90 inhibitors, the second-generation inhibitors have lower toxicity and improved bioavailability which support a higher treatment dosage.

The RD derivative NVP-AUY922 (luminespib or VER-2296) (Figure 4), a second-generation HSP90 inhibitor, overcomes the problem of instability in vivo and has shown anticancer efficiency in various animal models [49,50,51]. Another RD-derived agent, AT13387 (onalespib) (Figure 4), is reported to react potently against multiple tumor cell lines and, more importantly, showed a significant retention effect in non-small-cell lung cancer (NSCLC) [52]. Moreover, ganetespib (STA-9090) (Figure 4) is among the most anticipated second-generation inhibitors. Ganetespib is remarkably cytotoxic to tumor cells [53]. In addition, ganetespib displays sensitizing effects when combined with other anticancer therapies, such as radiotherapy [54,55].

It is worth mentioning that the discovery of the Bergerat fold provided clues for the development of more advanced second-generation HSP90 inhibitors—purine derivatives [56], including PU3, PU-H71 and MPC-3100 (Figure 5).

### 5.2. The C-Terminal Inhibitors

In multiple preclinical and clinical trials, the outcomes of N-terminal inhibitors were unsatisfactory due to their toxicity and drug resistance, which led to the development of C-terminal inhibitors. Novobiocin (NB) (Figure 6) is a C-terminal inhibitor isolated from Streptomyces species. NB binds to the C-terminus of HSP90 to disrupt the dimerization and induce the release of client proteins. NB depleted HSP90 client proteins such as HER2, v-sac and Raf-1 [57,58]. NB also interferes with the interaction of Hsp90 and the cochaperone HSP 70. Other C-terminal inhibitors, including clorobiocin, coumermycin A1 and epigallocatechin-3-gallate (EGCG) (Figure 6), have more effective activity and a lower therapeutic dosage.

## 6. Hsp90 Inhibitors as Potential Therapeutic Agents

There are several reasons that HSP90 inhibitors are ideal anticancer agents.

As mentioned above, the HSP90 clients contains a wide range of carcinogenic proteins, such as Akt, Bcr-Abl, C-Raf, B-Raf, EGFR, Her-2, HIF-1α, Met, VEGFR, and mutated p53. Most of these clients are key transducers in signaling pathways important for the development and progression of tumors. Hence, inhibiting HSP90 can simultaneously affect a wide range of client proteins, thereby shutting down multiple oncogenic signaling pathways.Cancer cells live in a much harsher environment. In addition to common pressures, such as hypoxia, low pH, and malnutrition, tumor cells face external pressures, including mutated client genes and proteins, altered signal pathways and the extra need for nutrients. Therefore, tumor cells depend more on HSP90 to maintain growth/survival than normal cells.As mentioned earlier, HSP90 is involved in many malignant behaviors, such as proliferation, metastasis/invasion, antiapoptosis, angiogenesis, and therapeutic resistance. Inhibition of HSP90 may simultaneously reverse multiple malignant behaviors of tumors.(Hsp90 often presents in a latent, uncomplex state in normal tissues. However, in tumor cells, Hsp90 entirely exists in multichaperone complexes with high ATPase activity and hence has high affinity for ATP and substrates. Thus, tumor cells are likely more sensitive to HSP90 inhibitors [59].

Unfortunately, to date, no HSP90 inhibitor has been approved by the FDA for the clinical treatment of cancer as a single agent. The first possible reason is the associated toxicities. Unacceptable adverse reactions have occurred in many preclinical and clinical trials. HSP90 is an indispensable molecular chaperone both in tumor cells and normal cells [1]. Once HSP90 is inhibited, basic cellular activity is greatly affected, followed by serious side effects. Liver and ocular toxicity are two common dose-limiting toxicities [60]. The main challenge is finding a therapeutic window with sufficient efficacy and low toxicity. Second, members of the HSP family form an enormous signaling network, and inhibiting a single HSP may lead to compensatory overexpression of other HSP members, such as HSP27 and HSP70 [61]. As a result, the inhibitory effects induced by HSP90 inhibition can be compensated. Moreover, the upregulation of other HSP molecules is believed to be associated with HSP90 inhibitor resistance. For example, upregulation of HSP27 was considered important for 17-AAG resistance through modulating glutathione (GSH) [62].

## 7. HSP90 Combination Therapy

Tumor heterogeneity plays an important role in tumor resistance to anticancer therapies. Anticancer treatments exert lethal pressures that trigger cancer cells’ adaptive behaviors. Cancer cells increase their survival abilities through rapid reactive mutations and chromosomal rearrangements. As tumors evolve and adapt, the effectiveness of anticancer treatments gradually diminishes. Radiotherapy and chemotherapy are still the two major anticancer therapies. Targeted and accurate therapies are being incorporated into cancer treatment. Unfortunately, therapeutic resistance can still emerge due to the tumors’ rapid adaptation processes. Resistance to radiotherapy and chemotherapy are the most common and troublesome phenomena in clinical treatment.

As stated above, HSP90 inhibitors can be considered ideal anticancer drugs. However, when used as monotherapy, HSP90 inhibitors did not achieve the desired effect. To date, no HSP90 inhibitor has been approved by the FDA for clinical monotherapy of cancer. Synergizing HSP90 inhibitors with other therapies appears to be a more plausible strategy. Many in vitro and/or in vivo studies have explored the combination of HSP inhibitors with other anticancer therapies. Most results suggest that HSP90 inhibitors have synergistic or additive effects with other anticancer therapies (Table 2). In addition, some ongoing or completed phase II and phase III clinical trials are overviewed in Table 3.

Hsp90 inhibitors can be considered sensitizers of cancer cells to other anticancer therapies. HSP90 inhibitors can enhance the efficacy of other anticancer treatments, such as radiotherapy and chemotherapy, by directly downregulating the pathways associated with drug resistance mechanisms and indirectly enhancing anticancer activity by inhibiting multiple tumor survival/growth pathways. In addition, combination therapy requires lower doses than monotherapy, thereby alleviating or even avoiding toxic side effects. We collected data from various studies on HSP90 inhibitor combination therapies and focused on the underlying mechanisms of each synergistic effect.

### 7.1. Chemotherapy

Taxanes are used as microtubule-targeting antitumor agents in cancer chemotherapy. Taxanes target β-tubulin in polymerized microtubules and cause mitotic arrest and apoptosis [80]. Preclinical data from different cancer cell lines and tumor xenograft models indicate that HSP90 inhibitors are synergistic with taxanes in targeting tumors. When 17-AAG and paclitaxel were combined, a significant growth inhibition effect of non-small-cell lung cancer (NSCLC) cells was observed in vitro and in vivo. The cytotoxicity of paclitaxel was enhanced 5–22-fold by 17-AAG [81]. In addition, in breast cancer with high levels of HER2 expression and amplification of AKT, Hsp90 inhibitor 17-AAG sensitized breast cancer cells to Taxol by causing the degradation of HER2 and the inactivation of Akt both in vitro and in vivo [32]. Some other studies can be seen in Table 2 [63,64].

Importantly, a randomized phase II study of ganetespib combined with docetaxel (GALAXY-1) was designed to evaluate efficacy and safety in advanced NSCLC. Although this study did not meet its primary endpoints, patients >6 months after the diagnosis of advanced lung adenocarcinoma showed significantly prolonged progression-free survival (PFS) and overall survival (OS) rates from this combination [82]. This finding led to the large-scale phase III trial of this combination (GALAXY-2) in patients with chemotherapy-sensitive advanced lung adenocarcinoma. Unfortunately, ganetespib synergizing with docetaxel did not improve survival in patients with advanced lung adenocarcinoma (NCT01798485) [83]. Subsequently, an article revealed the possible mechanisms for this failure. KRAS mutant NSCLC can rapidly obtain resistance to ganetespib due to bypass of ganetespib effects, inducing G2/M arrest. This acquired resistance to ganetespib then results in cross-resistance to docetaxel. Moreover, overactivated p90RSK-CDC25C signaling is the core of G2/M arrest. Administration of p90RSK inhibitors and/or CDC25C inhibitors may reverse KRAS mutant NSCLC resistance to ganetespib [84].

Another chemotherapy agent, cisplatin, is used to treat several cancer types, including sarcomas, carcinomas, lymphomas, and germ cell tumors. Cisplatin interacts with DNA bases and eventually causes apoptosis. 17-AAG demonstrated synergistic anticancer activity with cisplatin in pediatric solid tumor cells (neuroblastoma and osteosarcoma) by inducing depletion of IGF1R and AKT, which are two key antiapoptotic proteins [85]. In the case of cisplatin-resistant pancreatic ductal adenocarcinoma cells (PDACs), 17-AAG sensitizes PDACs to cisplatin. The underlying mechanism of this synergism is the degradation of Fanconi anemia pathway factors by 17-AAG. As a result, the repair of DNA adducts induced by cisplatin is eliminated [86,87]. A recent study showed that ganetespib combined with pemetrexed and cisplatin was safe and effective in patients with malignant pleural mesothelioma (MPM). The combination of antifolate and platinum is the first-line treatment for MPM. Ganetespib causes the degradation of the HSP90 client thymidylate synthase to reverse antifolate resistance [55]. Furthermore, a triple combination of SNX-5422, carboplatin, and paclitaxel followed by maintenance SNX-5422 therapy showed substantial tolerance and antitumor activity against NSCLC [88]. Other studies can be seen in Table 2 [65,66,67].

5-Fluorouracil (5-FU) is a nucleotide analog targeting thymidylate synthase (TS), a key enzyme in the de novo synthesis of dTMP. In a colorectal xenograft model, ganetespib showed synergistic effects with 5FU by inducing cell cycle arrest and downregulating the expression of thymidylate synthase [68]. In addition, ganetespib also inhibits multiple signaling pathways (PI3K/Akt, ERK) related to proliferation and therapeutic resistance [68].

### 7.2. Radiotherapy

The key role of HSP90 in the DNA damage response has been described above. HSP90 inhibitors can significantly induce depletion of essential HSP90 clients in the DNA damage response. In addition, an HSP90 inhibitor can abrogate S and G2/M cell cycle checkpoint controls by promoting the degradation of the client kinases CHK1 and WEE1 [89]. A quantitative spectrum analysis of protein expression changes after the administration of 17-DMAG revealed that DNA damage response pathways are among the most sensitive pathways at very low 17-DMAG concentrations [90]. This finding suggests that HSP90 inhibitors may selectively radiosensitize tumor cells.

HSP90 inhibitors have long been a promising solution to radiation resistance. Yin et al. reported that both BIIB021, an HSP90 inhibitor based on the purine scaffold, and 17-AAG showed anticancer effects on head and neck squamous cell carcinoma (HNSCC) xenografts as a single agent. However, xenografts treated with BIIB021 and radiation grew slower at the same time and even showed regression [91]. In a heterotopic transplantation model of colorectal cancer cells, the HSP90 inhibitor NW457 synergized with radiotherapy and induced a stronger inhibitory effect on tumor growth [42]. In another case of GBM, concurrent exposure to the HSP90 inhibitor NXD30001 and radiotherapy remarkably inhibited tumor growth and extended the median survival of tumor-bearing mice [92]. One study revealed that low, nontoxic doses of the Hsp90 inhibitor AT13387 (Onalespib) can selectively sensitize head and neck squamous cell carcinoma (HNSCC) and pancreatic cancer cells to radiotherapy, while there was no synergetic effect on normal cells [93]. Other studies can be seen in Table 2 [35,43].

### 7.3. Immunotherapy

Immune checkpoint blockade has received extensive attention recently due to its clinical activity in many types of human cancers. Response failures and adverse immune responses are major issues in the evolution of this immunotherapy. Some studies show that combining immune checkpoint blockade with HSP90 inhibitors may be a promising way to address these problems.

The HSP90 inhibitor ganetespib was found to potentiate the antitumor efficacy of the anti-PD-L1 antibody (STI-A1015) in mice bearing MC38 colon carcinoma tumors and B16 melanoma tumors. The mechanism contributing to this synergistic effect is that inhibition of HSP90 affects PD-L1 expression and HIF-1α, JAK2, and mutated EGFR [94]. Mbofung et al. reported that the HSP90 inhibitor ganetespib improved T-cell-mediated tumor cytotoxicity to melanoma cells. This effect of ganetespib could be explained by the upregulation of interferon response genes induced by ganetespib [95]. Moreover, in the MC-38 syngeneic mouse tumor model, ganetespib remarkably reduced the expression of immune checkpoint proteins, including PD-L1 and PD-L2 [96].

In a more recent study, combination treatment with the HSP90 inhibitor XL888 and PD-1 blockade was effective in pancreatic ductal adenocarcinoma (PDAC) models. PDAC is characterized by fibrotic stroma closely related to pancreatic stellate cells (PSCs) and cancer-associated fibroblasts (CAFs). XL888 downregulates IF-6 expression in PSCs/CAFs and directly inhibits PSC/CAF growth, thereby enhancing the efficacy of anti-PD-1 blockade [97]. Other studies can be seen in Table 2 [69,70,71].

### 7.4. Protein Kinase Inhibitors

Protein kinases are the largest single group of HSP90 clients. Several protein kinase inhibitors (PKIs) have been reported to synergize with Hsp90 inhibitors in killing tumor cells. Raf kinase is an HSP90 client; thus, HSP90 inhibition could promote the antitumor efficiency of Raf kinase inhibitors. When combining the Raf kinase inhibitor sorafenib with tanespimycin (17-AAG), clinical efficacy was observed in 9 of 12 renal cancer patients and 4 of 6 melanoma patients [98]. The addition of SCH727965 (SCH), a cyclin-dependent kinase inhibitor, to NVP-AUY922 (AUY) can induce apoptosis of osteosarcoma (OS) cells with no effect on normal osteoblasts or fibroblasts [99]. Another CDK inhibitor, dinaciclib, synergized with the novel HSP90 inhibitor HAA2020 and displayed stronger apoptotic and cell cycle control properties in acute myeloid leukemia (AML) [100].

BRAF and/or MEK inhibitors are important options for treating BRAFV600E mutant high-grade gliomas (HGGs). Nevertheless, therapeutic outcomes are often disappointing due to drug resistance. Recently, an in vitro and in vivo study reported that adding an HSP90 inhibitor may solve this problem. HSP90 inhibitors can deactivate the MAPK and AKT/mTOR pathways, which are reactivated by BRAF and/or MEK inhibitors. Therefore, HSP90 inhibitors combined with BRAF and/or MEK inhibitors induced apoptosis of HGG cells [101].

Approximately 20–30% of breast cancers are human epidermal growth factor 2 (HER2)-positive with more malignant characteristics than other breast cancer subtypes. HER2 is a member of the ErbB family of transmembrane receptor tyrosine kinases. The amplified HER2 gene induces overexpression of HER2, which leads to tumor proliferation, adhesion and aggression. HSP90 is known to modulate the tyrosine kinase activity of HER2. Once HSP90 is inhibited, HER2 cannot be folded properly and is eventually degraded. This degradation undoubtedly increases the efficacy of some HER2-targeted drugs, such as lapatinib and trastuzumab.

Lapatinib, a tyrosine kinase inhibitor inhibiting both HER2 and EGFR, has been widely used for treating HER2 (+) breast cancer patients. However, due to the acquired resistance of most patients, the prognosis is unsatisfactory. Lapatinib in combination with 17-DMAG showed a synergistic effect in suppressing cell proliferation in vitro and in vivo [102]. Another well-known kinase inhibitor targeting HER2, trastuzumab, is a humanized anti-HER2 antibody used to treat HER2 (+) breast cancer. Similar to lapatinib, most patients eventually develop resistance to the drug within 1–2 years. In a phase I clinical trial, combination therapy showed antitumor activity and great tolerance in patients with trastuzumab-resistant HER2 (+) breast cancers [103]. A phase II trial of 17-AAG plus trastuzumab in patients with HER2 (+) metastatic breast cancer revealed that this combination had a potent anticancer effect against those patients who previously progressed by using trastuzumab. Notably, this was the first phase II study to show the efficacy of 17-AAG, meeting Response Evaluation Criteria in Solid Tumors (RECIST) criteria in solid tumors [104]. Some other studies can be seen in Table 2 [72,73,74].

### 7.5. Proteasome Inhibitors

There is a complementary effect between HSP90 inhibitors and proteasome inhibitors. When both exist simultaneously, undegraded proteins will accumulate in cells, and consequent protective mechanisms induced by such accumulation will be prevented [105].

The proteasome inhibitor bortezomib in combination with 17-AAG was evaluated in patients with multiple myeloma (MM) in a phase I/II trial. The results showed that bortezomib plus 17-AAG was well tolerated, and bortezomib-naive patients had the highest response rates (41%) [106]. A subsequent phase III trial was suspended for nonclinical reasons.

The combination of the Hsp90 inhibitor KW-2478 with the proteasome inhibitor bortezomib showed a stronger inhibition of myeloma (MM) cell growth and synergistic antitumor efficacy in a subcutaneously inoculated human myeloma model [107]. Moreover, patients with relapsed/refractory MM showed substantial tolerance to this combination with no apparent overlapping toxicity in a phase I/II clinical trial. However, the antimyeloma activity of this combination was relatively modest [108]. Another HSP90 inhibitor, 17-DMAG, synergized with bortezomib at low toxic doses and showed therapeutic advantages in the growth inhibition of rhabdomyosarcoma (RMS) cells over single-agent treatment [109]. Other studies can be seen in Table 2 [75,76].

### 7.6. Histone Deacetylase Inhibitors

Histone deacetylases (HDACs) are responsible for the deacetylation of many proteins, including Hsp90. HDAC inhibitors can induce tumor cell apoptosis, growth arrest, cell cycle arrest, senescence, differentiation, and immunogenicity and inhibit angiogenesis via different downstream cellular pathways [110]. HDAC inhibitors displays great activity in hematological tumors. However, HDAC inhibitor’s outcomes with solid tumors are not that successful as expected. On the one hand, poor pharmacokinetics of many HDAC inhibitors prevent them from accumulating to effective concentrations in solid tumors as it does in blood tumors. One the other hand, low permeability can also affect the accumulation of HDAC inhibitors. As for hematologic tumors, HDAC inhibitors are easier to reach their therapeutic concentrations and a short half-life do not affect their anticancer activity [111].

AN effective HDAC inhibitor, Panobinostat (LBH589), is thought to exhibit best anticancer activity when combined with other therapies [112]. This suggest that combination therapy may be a great strategy to address HDAC inhibitors’ limited use. When an HDAC inhibitor is administered, HSP90 is hyperacetylated. Hyperacetylation can interfere with the function of HSP90 and ultimately lead to the degradation of oncogenic client proteins [113].

A dual effect inhibitor, MPT0G449, inhibits both HDAC and HSP90. In human acute leukemia, this dual inhibition leads to the downregulation of various oncogenic signaling pathways, including PI3K/AKT/mTOR, JAK/STAT, and RAF/MEK/ERK [114]. In Table 2 we collect other 2 studies [77,78].

### 7.7. Other HSP Inhibitors

Among the reasons for the limited efficacy of HSP90 inhibitor monotherapy is resistance to HSP90 inhibitors. When inhibition of HSP90 leads to downregulation of broad intracellular signal pathways, other members of the HSP family, such as HSP27 and HSP70, are upregulated for feedback, thereby neutralizing the effects induced by HSP90 inhibition [62,115]. The upregulation of HSP27 was reported to promote 17-AAG resistance by modulating glutathione (GSH) [62]. Several other studies have found this upregulation of HSP27 by HSP90 inhibition. In addition, overexpression of HSP70 can resist apoptosis induced by 17AAG [116].

Simultaneously, inhibiting several HSP molecules, including HSP90, may be the key to solving this issue. Inhibiting HSP 27 with Hsp27-specific siRNA was found to potentiate the inhibitory effect of the Hsp90 inhibitor GA on breast cancer stem-like cells (BCSCs) [115]. In another case, the HSP27 inhibitor OGX-427 enhanced the anticancer effects of the Hsp90 inhibitor PF-04929113 in castration-resistant prostate cancer (CRPC) xenografts [79]. Increased hsp70 levels are associated with antiapoptosis and limited efficacy of Hsp90-based treatments, which are realized through HSP70 inhibiting death receptor and mitochondria-initiated signaling for apoptosis. KNK437, an HSP70 inhibitor, reversed Hsp70-induced apoptosis and significantly enhanced the antileukemia activity of 17-AAG [116]. Therefore, other HSP inhibitors can be considered a solution to HSP90 inhibitor resistance in future clinical treatments.

### 7.8. Others

Hsp90 inhibitors also act synergistically with many other anticancer therapies, including photodynamic therapy (PDT) [117] and hormone therapy [118,119,120], via different mechanisms. Furthermore, some bifunctional drugs, such as STA-8666 (HSP90 inhibitor and topoisomerase inhibitor SN-38) [121], DHP1808 (HSP90-PI3K) [122] and MPT0G449, are essentially in combination. Generally, an Hsp90 inhibitor is a sensitizer of cancer cells to different therapies.

## 8. Conclusions and Future Decisions

In this review, we introduced the role of HSP90 in cancer development and HSP90 inhibitor therapeutic potential. As mentioned above, some issues with HSP90 inhibitors limit their clinical application as single agents. We discussed multiple combinational options with HSP90 inhibitors. The outcomes of these combination therapies suggest that combination therapy is a promising strategy. Taken together, based on various in vitro and in vivo experiments as well as clinical trials, we believe that kinase inhibitor and proteasome inhibitor may be more successful options. The former is because kinase account for a large proportion in HSP90’s clients. The latter is because proteasome inhibitor such as Bortezomib have entered phase III clinical trials which shows good prospects. In addition, the combination of HSP90 inhibitors plus other HSP inhibitors plus other anticancer therapies may be a great strategy to enhance HSP90 inhibitors’ effectiveness.

Among the classic mechanisms of HSP90 inhibitors is the induction of oncoprotein depletion. HSP90 currently has more than 700 proven client proteins; thus, more combinational options can be discovered in future clinical studies.

However, combination therapy effects are not entirely positive. The combination of two or more anticancer drugs may induce overlapping toxicity. For example, patients with advanced carcinomas and sarcomas were intolerant to the combination of the antiangiogenic agent ziv-aflibercept and the Hsp90 inhibitor ganetespib [117]. In the future design of novel combinations, multiple toxic reactions should be considered.

By analyzing various combination therapies, we found that cancers driven by HSP90-dependent oncogenic proteins are more sensitive to HSP90 inhibitors. Therefore, to target cancers more accurately, further study of the HSP90 client spectrum and molecular mechanisms of HSP90 is important. We also need more accurate patient stratification methods based on key HSP90 clients.

There are other promising directions that may address the current issues with HSP90 inhibitors and promote their clinical use. One direction is to develop more tumor-selective, less toxic HSP90 inhibitors. For instance, a hydrogen peroxide-activated Hsp90 inhibitor, Boro-BZide, was reported to selectively target cancer cells over normal cells. Special chemical modification of this agent can temporarily inactivate the parent HSP90 inhibitor and selectively release the HSP90 inhibitor in cancer cells after Boro-BZide is activated by hydrogen peroxide. Subsequent experiments showed that Boro-BZide had a significant suppressing effect against human breast cancer cells but not against normal breast cells [123]. Another direction is to develop Hsp90 isoformselective inhibitors. Grp94 has been reported to deeply associated with proliferation and metastasis of some types of cancers such as hepatocellular carcinoma [124], multiple myeloma [125] and inflammatory colon carcinomas [126]. Isoformselective inhibitors targeting Grp94 might be more effective for the cancers with high demand for proteins folding (multiple myeloma), inflammatory process (inflammatory colon carcinomas) and chronic infection (hepatocellular carcinoma) [127]. As for Trap-1, this mitochondrial isoform has association with tumor metabolism and mitochondrial homeostasis. Targeting Trap-1 might disrupt tumors’ metabolic reprogramming [15].

## Figures and Tables

**Figure 1 cells-11-02556-f001:**
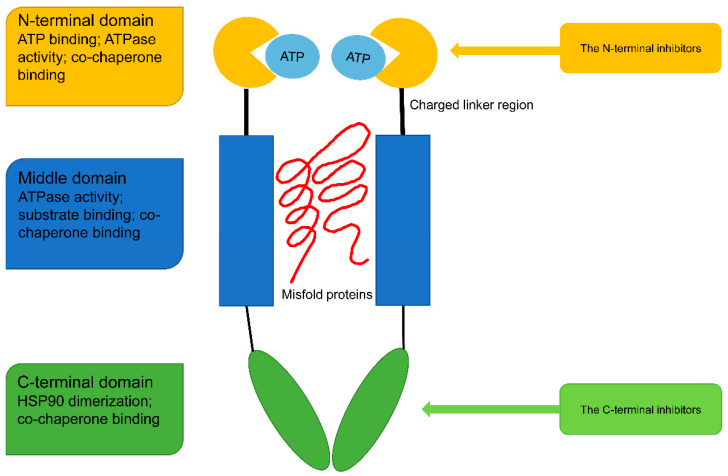
Basic structure of HSP90. The N-terminal domain contains ATP binding site with a special domain Bergerat fold; The charged linker region is a flexible linker between the N-terminal and middle domains; The middle domain is the binding site of client protein; The C-terminal domain is responsible for HSP90 dimerization; The N-terminal, middle domain and C-terminal domain all have the binding site of co-chaperones.

**Figure 2 cells-11-02556-f002:**
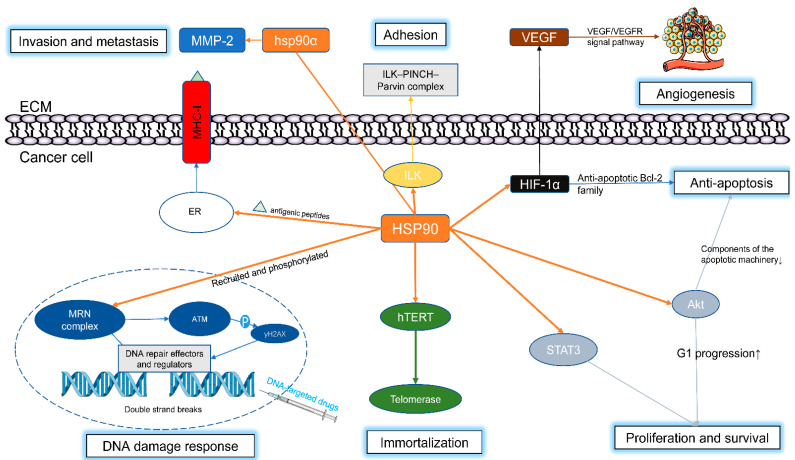
HSP90 in tumor progression. Many HSP90 client proteins are involved in numerous processes related to tumor progression. Hsp90 interacts with hTERT promoter, increases level of telomerase and promote immortalization of cancer cells; HSP90α is secreted into ECM and activate precursor of MMP which causes the decomposition of extracellular matrix; HSP90 client ILK recruits other components of ILK-PINCH-Parvin complex which is related to tumor’s adhesion; HIF-1α is closely related to HSP90. HIF-1α can affect expression level of VEGF which promote angiogenesis through VRGF/VEGFR pathway; HIF-1α enhances the antiapoptotic effect of cancer cells by the antiapoptotic BCL-2 family; HSP90 is recruited by the MRN complex to DNA damage site and phosphorylated by MRN. HEP90 assist MRN to activate ATM and initiate a series pathways of DNA damage repair; HSP90 client Akt can promote antiapoptotic effect of cancer cells by downregulating components of the apoptotic machinery. Akt promote tumor growth by accelerating G1 progression. HSP90 escorts the antigenic peptides to the ER and the antigenic peptides are loaded onto the newly synthesized MHC-I molecules. The MHC-I molecules carrying antigenic peptides are transported to the surface of cancer cell.

**Figure 3 cells-11-02556-f003:**
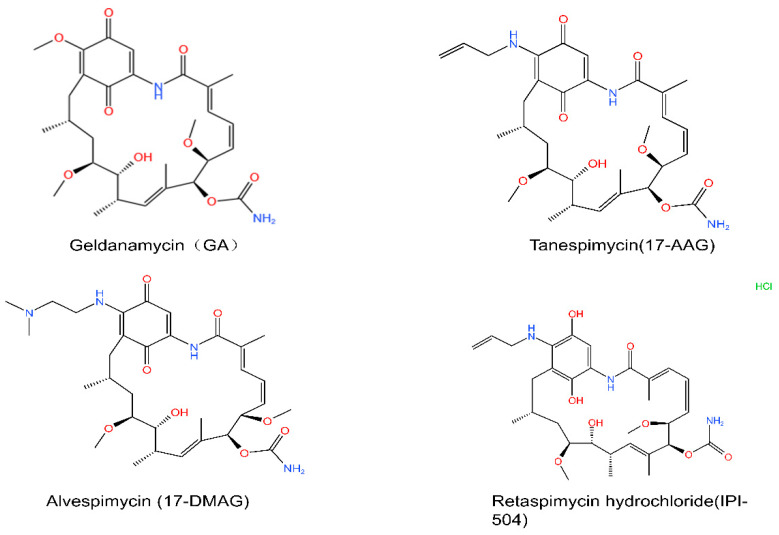
Chemical structures of GA-derived HSP90 inhibitors. GA and its derivatives 17-AAG, 17-DMAG and IPI-504. Chemical structures are from http://www.chemspider.com (Accessed on 16 August 2022).

**Figure 4 cells-11-02556-f004:**
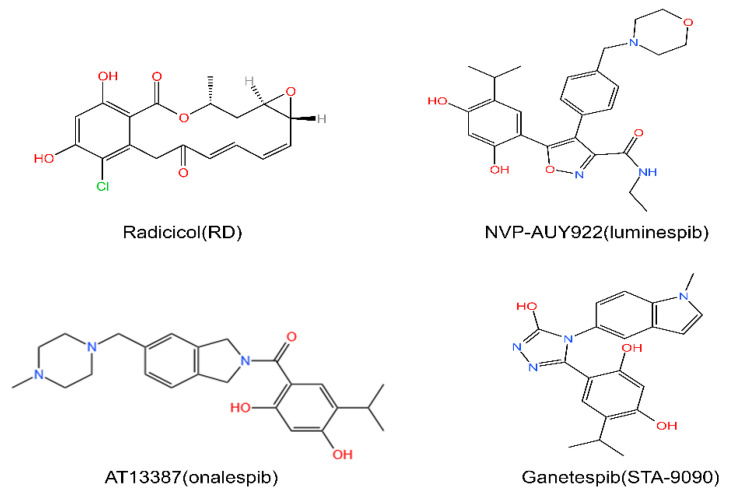
Chemical structures of RD-derived HSP90 inhibitors. RD and its derivatives NVP-AUY922, AT13387, Ganetespib. Chemical structures are from http://www.chemspider.com (Accessed on 16 August 2022).

**Figure 5 cells-11-02556-f005:**
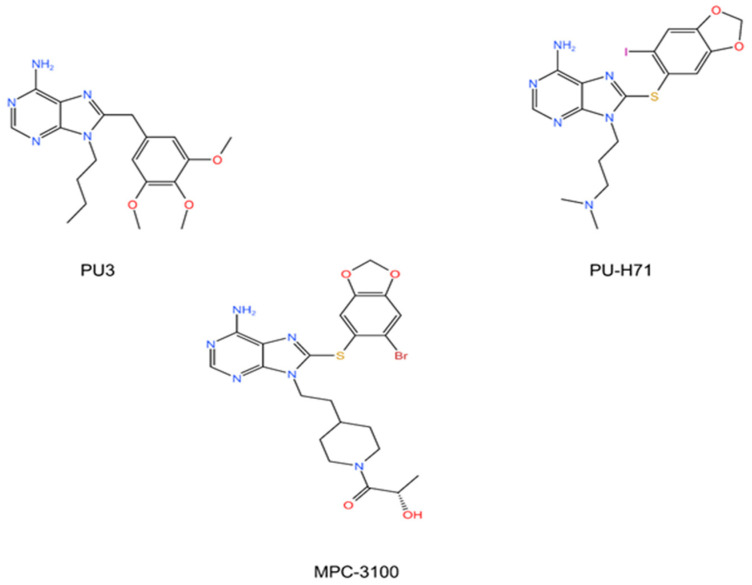
Purine and purine-like inhibitors of HSP90. PU3, PU-H71 and MPC-3100RD. Chemical structures are from http://www.chemspider.com (Accessed on 16 August 2022).

**Figure 6 cells-11-02556-f006:**
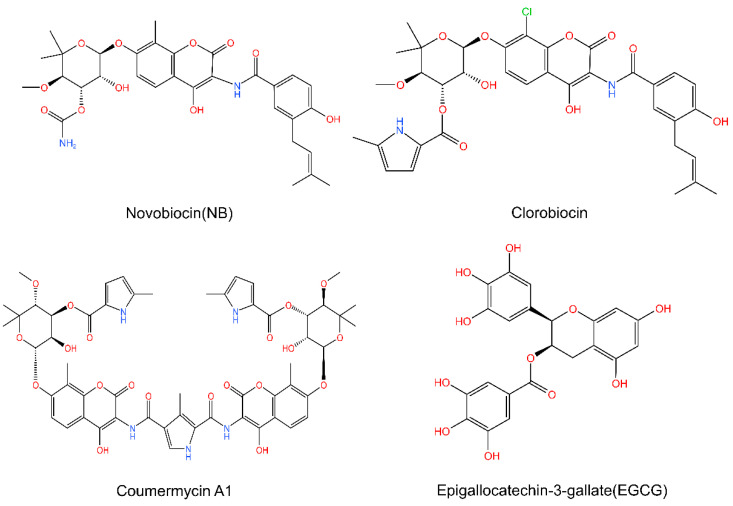
The C-terminal inhibitors. Novobiocin, clorobiocin, coumermycin A1 and epigallocatechin-3-gallate (EGCG). Chemical structures are from http://www.chemspider.com (Accessed on 16 August 2022).

**Table 1 cells-11-02556-t001:** HSP90’s oncogenic clients.

Client	Function	Cancer
Transcription factors		
p53	Tumor-suppressor protein	Mutated in cancer
HIF-1α	Heat-shock response	Angiogenesis
Estrogen receptors	Response to estrogens	Breast cancer
Androgen receptor	Response to androgens	Prostate cancer [20]
OCT4	Embryonic development and induction of pluripotent cells	Chemotherapy resistance and tumor differentiation [21]
STAT3	JAK–STAT3 signal pathway	Proliferation and survival [22]
Kinase		
AKT (PKB)	PI3K/AKT pathway	impaired apoptosis
CDK4(Cyclin D-dependent kinases 4)	Cell cycling	Tumor proliferation [23]
SRC	Nonreceptor tyrosine kinase	Tumor growth and metastasis [24]
BRAF	Mitogen signaling	Proliferation and invasion
JAK (Janus kinases)	JAK–STAT signal pathway	Proliferation and survival
BCR–ABL	Fusion tyrosine kinase	Hallmarks of CML cells
HCK (hematopoietic cell kinase)	Immune response	Distant metastasis
IκB	Activation of NF-κB pathway	Proliferation, antiapoptotic and angiogenesis [25]
MMP2 (matrix metalloproteinase 2)	Decomposition of extracellular matrix (ECM) components and basement membrane (BM)	Invasion/metastasis
Others		
TERT	Telomere maintenance	Immortalization
RAD51 and/or RAD52	DNA repair	Radiotherapy resistance

**Table 2 cells-11-02556-t002:** HSP90 inhibitors synergizing with other anticancer therapy.

Other Anticancer Treatment	Hsp90 Inhibitor	Cancer Cell Type	Synergistic Mechanism	Conditions	Refs
Chemotherapy					
Taxanes	17-AAG	EGFR mutant non–small-cell lung cancers (NSCLC)	Degradation of epidermal growth factor receptor (EGFR)	In vitro, in vivo	[63]
	Ganetespib, NVP-AUY922	Triple-negative breast cancer (TNBC)	Degradation of Glucocorticoid receptor (GR), apoptosis↑proliferation↓	In vitro, in vivo	[64]
Cisplatin	17-AAG	Cisplatin-resistant esophageal squamous cell carcinoma (ESCC)	Apoptosis↑by Akt/XIAP pathway	In vitro	[65]
	17AAG, ganetespib	Platinum-resistant ovarian cancer	Apoptosis↑, DNA damage↑	In vitro, in vivo	[66]
	17AAG	Relapsed diffuse large B-Cell lymphoma (DLBCL)	Apoptosis↑, DNA damage↑	In vitro	[67]
5-flfluorouracil (5-FU)	ganetespib	Colorectal cancer (CRC)	Inducing G0/G1 cell cycle arrest; downregulating thymidylate synthase	In vitro, in vivo	[68]
Radiotherapy					
Radiotherapy	Ganetespib	Pancreatic ductal adenocarcinoma (PDAC)	Proliferation↓, angiogenesis↓, apoptosis↑, HIF-1α expression↓	In vitro, in vivo	[35]
Fractionated, conebeam CT (CBCT)-based irradiation	NW457	Glioblastoma	Disrupting DNA damage response (DDR)	In vitro, in vivo	[43]
Immunotherapy					
Cellular immunotherapy	17-AAG	Wild-type BRAF, NRAS mutant melanoma cells	ERK signaling↓, CRAF↓	In vitro, in vivo	[69]
Anti-PD-1 antibody	Ganetespib	PDAC	Downregulating STAT1; indoleamine 2,3-dioxygenase 1 (IDO1) ↓, PD-L1↓	In vitro, in vivo	[70,71]
Protein kinase inhibitors					
mTOR inhibitor AZD8055	AUY922	breast cancer	Enhancing cell cycle arrest; destabilizing multiple tyrosine kinases; abrogating activation of AKT induced by AZD8055	In vitro, in vivo	[72]
EGFR inhibitor (erlotinib, gefitinib)	Ganetespib	NSCLC	Stabilizing EGFR protein levels in an inactive state; completely abrogating ERK and AKT signaling activity	In vitro, in vivo	[73]
MET kinase inhibitor Crizotinib	Ganetespib	MET-driven cancers	Synergistically inhibiting MET and its downstream signaling pathways	In vitro, in vivo	[74]
Proteasome inhibitors					
Bortezomib	IPI-504	Mantle cell lymphoma (MCL)	Inducing depletion of BiP/Grp78, inhibiting unfolded protein response, promoting NOXA-mediated mitochondrial depolarization	In vitro, in vivo	[75]
Bortezomib	PU-H71	Ewing sarcoma	G2/M phase arrest; depletion of proteins including AKT, pERK, RAF-1, c-MYC, c-KIT, IGF1R, hTERT and EWS-FLI1	In vitro, in vivo	[76]
Histone deacetylases inhibitors					
LBH589	17-AAG	CML, AML	Degradation of FLT-3 and Bcr-Abl↑	In vitro	[77]
PXD101, suberoylanilide hydroxamic acid (SAHA), trichostatin A (TSA)	SNX5422	Anaplastic thyroid carcinoma (ATC)	Inducing cell death by suppressing PI3K/Akt/mTOR signaling	In vitro	[78]
Other HSP inhibitors					
HSP27 inhibitor OGX-427	17-AAG	Castration-resistant prostate cancer (CRPC)	OGX-427 attenuates Hsp27 expression induced by HSP90 inhibitor; ER stress↑ apoptosis↑	In vitro, in vivo	[79]
Other therapies					
Fulvestrant (hormone therapy)	AUY922	ER positive breast cancer	Downregulation of ErbB receptors and downstream PI3K/AKT and ERK pathway; reversing Fulvestrant resistance	In vitro	[79]

**Table 3 cells-11-02556-t003:** Overview of HSP90 inhibitors combination therapy in clinical trial (Home-ClinicalTrials.gov).

Anticancer Therapy	Hsp90 Inhibitor	Cancer Type	Phase	Status	NCT Number
Abiraterone acetate	AT13387	Prostate Cancer	1/2	Completed	NCT01685268
Crizotinib	AT13387	Non-small-cell lung cancer (NSCLC)	1/2	Completed	NCT01712217
Erlotinib hydrochloride	AUY922	Non-small-cell lung cancer (NSCLC)	1/2	Completed	NCT01259089
Niraparib, carboplatin	Ganetespib	Ovarian Cancer	2	Active, not recruiting	NCT03783949
Paclitaxel	Ganetespib	Epithelial ovarian cancer (EOC)	2	Terminated	NCT02012192
Fulvestrant	Ganetespib	HR+ breast cancer	2	Completed	NCT01560416
Paclitaxel	Ganetespib	Recurrent fallopian tube cancer; Recurrent ovarian epithelial cancer; Recurrent primary peritoneal cavity cancer	1/2	Terminated	NCT01962948
Trastuzumab	AUY922	Advanced gastric cancer	2	Terminated	NCT01402401
Trastuzumab	AUY922	Advanced HER2-positive breast cancer	1/2	Completed	NCT01271920
Bortezomib	KW-2478	Multiple Myeloma	1/2	Completed	NCT01063907
Sirolimus	Ganetespib	Malignant peripheral nerve sheath tumors (MPNST); Sarcoma	1/2	Completed	NCT02008877
Bortezomib	AUY922	Relapsed or refractory multiple myeloma	1/2	Completed	NCT00708292
Bortezomib	17-AAG	Multiple Myeloma	2/3	Completed	NCT00514371
Bortezomib	17-AAG	Multiple Myeloma	3	Completed	NCT00546780

## Data Availability

Not applicable.

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
