# Peer review of "Targeting Heat-Shock Protein 90 in Cancer: An Update on Combination Therapy"

_cells, 2022, doi:10.3390/cells11162556_

Round 1

Reviewer 1 Report

This review summarizes the use of HSP90 inhibitors for the treatment of cancers. Overall, it is well written with clear structure. However, there are some issues that should be addressed, including repetition of some contents, lack of supporting references, the excessive use of abbreviation (which makes the article reader unfriendly), etc. It would be helpful if the authors can discuss which of the combination therapy is likely to be more successful. It is also interesting that only two of the four authors contributed toward writing of this review. Some specific points are listed below:

Need ref for the first paragraph, line 40, line 91-98, line 103, Table 1, line 126-132, line 171-175, line 207, line 227, line 284, Table 2/3,  .

I can not access the ref "Hsp90facts.pdf (picard.ch)".

Some contents in paragraphs beginning at line 48 and 64 are repetitive. Please also specifiy if the charged linker region is of positive or negative charges.

I am confused by the paragraph of line 119. Do the author mean only the mutant but not the wild-type proteins are the clients of HSP90? Or both wild-type and mutant proteins are the clients?

Line 140 should refer to Figrue 2?

Please explain the difference of second-generation HSP90 inhibitors from the first generation.

Change the paragraph of line 265 into bullet points.

Please define the difference of "clientele" and "clients". If they are interchangeable, please chose one and unify the usage.

Figure resolution seems to be quite low.

Please include the chemical strucutre of the inhibitors.

Author Response

Thanks very much for taking your time to review this manuscript.

Point 1: The excessive use of abbreviation

Response 1: We thank the reviewer for this suggestion. We have added full names for some important terminologies. You can see them in line 102, line 107, line 148, line 156.

Point 2: It would be helpful if the authors can discuss which of the combination therapy is likely to be more successful.

Response 2: We thank the reviewer for the careful evaluation. In the conclusion section, we provide the combinations that are relatively more successful according to our understanding. Please see our revised manuscript and they are in line 540-546.

Point 3: It is also interesting that only two of the four authors contributed toward writing of this review

Response 3: We thank the reviewer for the careful evaluation. We have already communicated with our assigned editor from Cells and this mistake has been fixed.

Point 4: Need ref for the first paragraph, line 40, line 91-98, line 103, Table 1, line 126-132, line 171-175, line 207, line 227, line 284, Table 2/3,

Response 4: We thank the reviewer for the careful evaluation. We’ve added references of the mentioned paragraphs. Please see our revised manuscript.

Point 5: I can not access the ref "Hsp90facts.pdf (picard.ch)".

Response 5: We've updated the link in the revised manuscript.

Point 6: Some contents in paragraphs beginning at line 48 and 64 are repetitive. Please also specifiy if the charged linker region is of positive or negative charges.

Response 6: We thank the reviewer for this suggestion. The repetitive paragraphs have been revised. In addition, we’ve added the charge properties of the charged linker.

Point 7: I am confused by the paragraph of line 119. Do the author mean only the mutant but not the wild-type proteins are the clients of HSP90? Or both wild-type and mutant proteins are the clients?

Response 7: We thank the reviewer for these comments. We will be more precise about this paragraph. Both wild-type and mutant proteins are the clients of HSP90. Our intention is that some mutant gene proteins are still under HSP90’ modulation even in mutant forms. You will see the revised version in our manuscript.

Point 8: Line 140 should refer to Figure 2?

Response 8: We have corrected this error in the manuscript.

Point 9: Please explain the difference of second-generation HSP90 inhibitors from the first generation.

Response 9: We’ve discussed the difference of second-generation HSP90 inhibitors from the first generation in our revised manuscript. Please see paragraph 248-252.

Point 10: Change the paragraph of line 265 into bullet points

Response 10: We’ve adjusted the paragraph as requested.

Point 11: Please define the difference of "clientele" and "clients". If they are interchangeable, please chose one and unify the usage.

Response 11: We decide to uniformly adopt the usage of "clients".

Point 12: Figure resolution seems to be quite low.

Response 12: We replace the figures with higher resolution.

Point 13: Please include the chemical strucutre of the inhibitors.

Response 13: We have added 4 figures including the chemical structure of the inhibitors. Figure 3-6.

Reviewer 2 Report

The review by Ren et al. focuses on the targeting of Heat Shock Protein 90 for the treatment of cancer. The report is well written and undoubtedly interesting, as cytoprotective proteins are considered to play a key role in the development of several cancers. Notwithstanding, it would be useful to provide information on the difrerent classes of heat shock proteins in the introduction, especially since they briefly mention them elsewhere in the report.

Author Response

Thanks very much for taking your time to review this manuscript.

Point1: it would be useful to provide information on the difrerent classes of heat shock proteins in the introduction, especially since they briefly mention them elsewhere in the report.

Response 1: We thank the reviewer for the careful evaluation. We have provided information about other members of HSP family at the beginning of the introduction. Please see our manuscript.